# EGCG Alleviates Oxidative Stress and Inhibits Aflatoxin B_1_ Biosynthesis via MAPK Signaling Pathway

**DOI:** 10.3390/toxins13100693

**Published:** 2021-09-30

**Authors:** Dan Xu, Shurui Peng, Rui Guo, Lishan Yao, Haizhen Mo, Hongbo Li, Hongxin Song, Liangbin Hu

**Affiliations:** School of Food and Biological Engineering, Shaanxi University of Science and Technology, Xi’an 710021, China; xudan@sust.edu.cn (D.X.); pshurui@126.com (S.P.); rui.guo@bioraypharm.com (R.G.); lisa_yaoo@126.com (L.Y.); mohz@sust.edu.cn (H.M.); hongbo715@163.com (H.L.); songhx@sust.edu.cn (H.S.)

**Keywords:** EGCG, *Aspergillus flavus*, AFB_1_, oxidative stress, MAPK signaling pathway

## Abstract

Aflatoxin biosynthesis has established a connection with oxidative stress, suggesting a prevention strategy for aflatoxin contamination via reactive oxygen species (ROS) removal. Epigallocatechin gallate (EGCG) is one of the most active and the richest molecules in green tea with well-known antioxidant effects. Here, we found EGCG could inhibit aflatoxin B_1_ (AFB_1_) biosynthesis without affecting mycelial growth in *Aspergillus flavus*, and the arrest occurred before the synthesis of toxin intermediate metabolites. Further RNA-seq analysis indicated that multiple genes involved in AFB_1_ biosynthesis were down-regulated. In addition, EGCG exposure facilitated the significantly decreased expression of AtfA which is a bZIP (basic leucine zipper) transcription factor mediating oxidative stress. Notably, KEGG (Kyoto Encyclopedia of Genes and Genomes) analysis indicated that the MAPK signaling pathway target transcription factor was down-regulated by 1 mg/mL EGCG. Further Western blot analysis showed 1 mg/mL EGCG could decrease the levels of phosphorylated SakA in both the cytoplasm and nucleus. Taken together, these data evidently supported that EGCG inhibited AFB_1_ biosynthesis and alleviated oxidative stress via MAPK signaling pathway. Finally, we evaluated AFB_1_ contamination in soy sauce fermentation and found that EGCG could completely control AFB_1_ contamination at 8 mg/mL. Conclusively, our results supported the potential use of EGCG as a natural agent to prevent AFB_1_ contamination in fermentation industry.

## 1. Introduction

Aflatoxins (AFs), the most carcinogenic and toxic naturally occurring metabolites, are primarily produced by *Aspergillus parasiticus* and *Aspergillus flavus* [1,2,3]. They were widely distributed in food and feed crops, which caused huge losses in crop quality and posed a global threat to animal and human health through the food chains [4,5,6]. Among them, aflatoxin B_1_ (AFB_1_) is categorized as Group I carcinogen by IARC (International Agency for Research) and induced as many as 28% of liver cancer cases in the world [7,8,9]. Hence, it is extremely urgent to explore simple and effective methods to prevent the contamination of AFs in feed and food, especially during processing and storage [10].

Oxidative stress occurs when ROS (reactive oxygen species) accumulation surpasses the intracellular antioxidant ability [11,12]. It has been implicated as a prerequisite for AFs formation [13,14]. Studies have shown that compounds associated with oxidative stress also influence the production of AFs in *A. flavus* [15,16]. For example, the antioxidant, ethylene, could inhibit aflatoxin biosynthesis in *A. flavus* by reducing oxidative stress and regulating the state of glutathione redox [17]. Gallic Acid inhibited aflatoxin formation by reducing oxidative stress and modulating transcription factors FarB and CreA [18]. Piperine could suppress AFB_1_ synthesis by regulating oxidative stress response in *A. flavus* [19].

As the richest catechin extracted from green tea, epigallocatechin gallate (EGCG) is reported to contain the highest concentration of powerful antioxidants [20,21]. Previous studies have shown that tea extract could suppress aflatoxin production by reducing the expression of *aflS* and *aflR* [22]. Green tea polyphenols (GTPs) could modulate AFB_1_ metabolism [23,24]. However, it has not been clarified whether EGCG in GTPs is responsible for the AFs suppression. In the present study, the functions of EGCG on fungal growth and AFB_1_ biosynthesis were explored. Further, RNA-seq and Western blot were conducted to elucidate the anti-aflatoxigenic mechanism of EGCG. Our study provided an efficient and safe agent to prevent the AFB_1_ contamination during storage and processing.

## 2. Results

### 2.1. EGCG Inhibiting AFB_1_ Synthesis, but Not Mycelial Growth

In order to investigate the function of EGCG in *A. flavus*, different doses of EGCG were set to determine their influence on mycelial growth and AFB_1_ production. As shown in Figure 1A, there was almost no effect on mycelia growth under the treatment of 0–2 mg/mL EGCG. Nevertheless, EGCG showed significant anti-aflatoxigenic properties. The results indicated that EGCG arrested the biosynthesis of AFB_1_ in a dose-dependent manner (Figure 1B). On treatment with 2 mg/mL EGCG, the inhibitory percentage of AFB_1_ biosynthesis was 92%. After confirming that EGCG at concentrations of up to 1 mg/mL could inhibit AFB_1_ biosynthesis and had no effect on mycelial growth, the time dependent was studied between 1 mg/mL EGCG and AFB_1_ biosynthesis. The results showed the inhibitory effect of EGCG on AFB_1_ production was hardly affected at different time points and the inhibitory rate still lived up to 87.5% at 8 days (Figure 1C). These results revealed that EGCG could inhibited AFB_1_ biosynthesis without affecting mycelial growth.

### 2.2. EGCG Blocking ROS Accumulation

EGCG is an ideal natural antioxidant. It is well known that fungal oxidative stress is associated with the AFs production [14]. Considering the functions of ROS regulating AFB_1_ biosynthesis, we managed to detect the intracellular levels of ROS through fluorescence microscopy with a fluorescent probe of DCFH-DA. The results showed that the generation of ROS in *A. flavus* cells was obviously decreased under the treatment of 1 mg/mL EGCG and the inhibition rate up to 60.55% at 48 h (Figure 2A). Through microscope observation, the mycelial tip with 1 mg/mL EGCG showed an extremely weak green fluorescence before 60 h (Figure 2B). In summary, the results revealed that EGCG could effectively inhibit the accumulation of ROS and relieve oxidative stress.

### 2.3. Transcriptome Analysis and qRT-PCR

To further understand the inhibitory roles of EGCG to AFB_1_ production, the RNA profile mycelia exposed to EGCG was sequenced and analyzed. In comparison to the control, a total of 927 differentially expressed genes (DEGs) were screened, including 496 markedly down-regulated genes and 431 significantly up-regulated genes (Figure 3A). Gene ontology (GO) analysis revealed a significant enrichment of DEGs in several functional categories. These down-regulated genes mainly involve the transcriptional regulation from RNA polymerase II promoter under oxidative stress, ascospore formation, signaling pathway via G-protein coupled receptor, inhibition of MAPK activity in osmosensory pathway, MAPK cascade, etc. (Figure 3B, Table 1). KEGG pathway enrichment analysis (Figure 3C, Table 2) revealed the synergistic relationship between these DEGs; the hypergeometric Fisher test was used to perform significant statistics on the pathways where DEGs were involved, revealing 15 enrichment pathways. Among these 15 pathways, down-regulated DEGs in the pathways of MAPK signaling pathway ranked first.

The transcription factors of basic leucine zipper (bZIP) respond to oxidative stress. Transcriptome analysis showed a great number of genes involved in bZIPs were down-regulated. Among them, AtfA was the most obvious one and qRT-PCR showed it was reduced by 51% after treatment with 1 mg/mL EGCG. It indicated that the inhibition of AFB_1_ synthesis by EGCG was related to the alleviation of oxidative stress.

To investigate the inhibitory mechanism of EGCG to AFB_1_ production, DEGs in the AFB_1_ biosynthesis cluster in *A. flavus* were identified based on transcriptomic data. It was indicated that five DEGs involving AFB_1_ biosynthesis were down-regulated. Furthermore, qRT-PCR was performed to validate these five DEGs, and the results displayed that the transcription of *aflE*, *aflC* and *aflA* was reduced by 60, 40 and 59% when *A. flavus* was treated with EGCG (Figure 3D, Table 3).

### 2.4. EGCG Arresting SakA Phosphorylation in MAPK Signaling Pathway

The signaling pathway via MAPK plays a vital role in fungal response to oxidative stress [25]. RNA-seq results suggested anti-oxidative stress system might be activated through the MAPK signaling pathway. To confirm our hypothesis whether the protective effects of EGCG against the reduction of oxidative stress is associated with the MAPK signaling pathway, we estimated the protein level of SakA by Western blot. It was indicated that EGCG did not affect the expression of SakA, but significantly decreased the levels of phosphorylated SakA in cytoplasm. The grayscale value of the protein bands analyzed by Image J showed that the inhibition rate of EGCG on SakA phosphorylation in cytoplasm was 38.92% (Figure 4A). AtfA mainly locates in the nucleus, whereas SakA translocates to the nucleus answering oxidative stress signals and reacts with AtfA [26]. Therefore, we examined the p-SakA in the nucleus (Figure 4B). EGCG (1 mg/mL) was identified to significantly reduce p-SakA with the inhibitory rate of 78.64% compared with the control. These results demonstrated that EGCG might alleviate oxidative stress through the MAPK pathway.

### 2.5. EGCG Inhibiting NOR Production

RNA-seq results showed *aflA* and *aflC* involved in NOR production were down-regulated under the treatment of 1 mg/mL EGCG. As a stable aflatoxin precursor, NOR is the first to be generated in AFB_1_ biosynthesis [27]. Therefore, NOR was detected to further analyze if EGCG inhibited NOR biosynthesis. Spectrophotometric analyses indicated that NOR biosynthesis were markedly suppressed by 1 mg/mL EGCG (Figure 5). The inhibition rate of NOR synthesis was only 0.45% of that in control group exposure to EGCG for 5 d. These results indicated that the AFB_1_ biosynthesis pathway was blocked prior to the production of NOR by EGCG.

### 2.6. Preventing AFB_1_ Production in Fermented Soybean Products with EGCG

The filamentous fungus *Aspergillus oryzae* plays a necessary role in the traditional fermentation and food processing industries and is often used in the production of soy sauce. Whereas it may be contaminated by *Aspergillus flavus* producing AFB_1_ during fermentation. As shown in Figure 6, EGCG showed significant inhibitory effects on AFB_1_ production in fermented soybean products. When 8 mg/mL EGCG was applied, the production of AFB_1_ was completely inhibited. Therefore, EGCG had great potential to control AFB_1_ contamination in fermented soybean products.

## 3. Discussion

Among the polyphenols in green tea extract, EGCG is the most active, accounting for 50%–80% of the catechin content [28]. Extensive studies have shown that it exhibits a wide range of beneficial effects including anti-oxidative, antimicrobial and anti-carcinogenic and has therapeutic potential against various human diseases [29]. In this study, we found that EGCG was able to dose-dependently inhibit AFB_1_ synthesis in *A. flavus*. The transcription files indicated that some genes responsible for AFB_1_ biosynthesis were down-regulated after the treatment of EGCG. Aflatoxins are biosynthesized by a polyketide pathway which involves at least 27 enzymatic reactions [7]. It was worth noting that EGCG targeted the genes responsible for the initial steps of AFB_1_ biosynthesis pathway, of which *aflA* and *aflC* were significantly down-regulated post the addition of EGCG. *aflA* and *aflC* encode fatty acid synthases and polyketide synthase, respectively [2]. They are also required for generation of NOR, a precursor of AFB_1_ and toxic intermediate metabolites. As we expected, no NOR accumulation occurred when EGCG blocked AFB_1_ biosynthesis. This highlighted the potential of EGCG as a natural material to prevent AFB_1_ contamination.

Oxidative stress is considered as a prerequisite for the synthesis of AFB_1_ in *A. flavus* [30,31]. bZIPs play crucial roles in pathways of stress response and are highly conserved in eukaryotes [32]. In filamentous fungi, many bZIP elements such as AtfA, AtfB, AP-1, SrrA, MeaB and MsnA respond to oxidative stress, and mediate the coordinative adjustment in the morphological development and secondary metabolism involving the AFB_1_ biosynthesis [33,34]. In the present study, the level of intracellular ROS in *A. flavus* was much reduced with the addition of EGCG, and AtfA associated with oxidative stress was also significantly down-regulated. Hence, we proposed that EGCG block AFB_1_ biosynthesis pathway by relieving oxidative stress. It has been indicated that cinnamaldehyde and ethanol could inhibit AFB_1_ formation by down-regulating the expression of some oxidative stress-related genes, whereas they also disrupt cell membranes [15,16]. EGCG is highlighted for its ability to pass through cell membrane [35], and hence has high potential as an intracellular antioxidant applied in food fermentation industry or health care.

How does oxidative stress affect transcription factor AtfA by EGCG? Transcription files revealed that MAPK signaling pathway was blocked after EGCG addition. Being a critical phosphorylation pathway, the modules of MAPK cascade converts environmental stimuli into cellular response [36,37]. They harmonize miscellaneous cellular processes, including development, signal transduction, and secondary metabolism [38,39]. In *A. nidulans*, SskA, as a response regulator, transmitted the signals of oxidative stress to SakA, Then, SakA translocated to the nucleus answering oxidative stress signals and reacted with AtfA [26]. Garrido-Bazán [40] has shown that H_2_O_2_ induced SakA accumulation in the nucleus, interacting with AtfA in *A. nidulans*. In *A. fumigatus*, SakA played a collaborative role in translating stress tolerance to the conidia through AtfA [40]. Further Western blot analysis found phosphorylation of SakA was decreased treated with EGCG both in cytoplasm and nucleus. Taken together, we proposed a hypothetical mechanism of EGCG underlying its inhibitory effects on AFB_1_ production (Figure 7). EGCG perturbed MAPK signaling pathway by decreasing the level of intracellular ROS. Simultaneously, the decrease of phosphorylated SakA level might reduce the tolerance to oxidative stress, leading to the activation of bZIP transcription factor AtfA in the stress signaling pathway. The decreased expression of genes in AFB_1_ cluster might then be the most down-stream response to the modulation by bZIP transcription factor.

Soy sauce is a traditional condiment fermented by *A. oryzae*. It is subject to *A. flavus* contamination producing AFB_1_ in the process of brewing soy sauce. In this case, it is required to develop efficient agents to prevent AFB_1_ contamination without affecting the growth of *A. oryzae*. Our study indicated that EGCG was able to block the pathway of AFB_1_ biosynthesis prior to NOR production without affecting mycelial growth. Given the safety and efficacy of EGCG, as well as its low cost, it could be considered as a promising agent to prevent the AFB_1_ contamination in fermentation industry.

## 4. Conclusions

In this study, we discovered that EGCG had high potential to reduce AFB_1_ biosynthesis in *A. flavus*. The anti-aflatoxigenic property of EGCG was related to its ability to relieve oxidative stress via MAPK signaling pathway and inhibit AFB_1_ cluster gene expression. In summary, our findings indicated EGCG could be considered as a potential natural agent to prevent AFB_1_ contamination.

## 5. Materials and Methods

### 5.1. Chemicals and Strain

EGCG of 98% purity, was purchased from Waston Technology Co., Ltd. (Shaanxi, China). *A. flavus* NRRL 3357 was gifted by Prof. Zhumei He of Sun Yat-sen University (Guangdong, China), and cultivated on the potato dextrose agar (PDA) medium (Becton, Dickinson and Company, Franklin Lakes, NJ, USA) at 30°C in dark. Spores were harvested with 0.05% TritonX-80 and their numbers were counted with a hemocytometer.

### 5.2. Determination of Fungal Growth and AFB_1_ Biosynthesis

Spores (10^6^ cells/mL) were inoculated into 20 mL YES containing 0, 0.25, 0.5, 1, and 2 mg/mL EGCG, respectively. All inocula were incubated at 28°C, 150 rpm. After incubation for 5 days, mycelia and filtrate were harvested. The dry weight of mycelia was weighed 12 h post desiccation at 80 °C.

The amount of AFB_1_ in filtrate was detected according to Dan et al. [41] with minor changes. AFB_1_ in the filtrate sample (1 mL) was extracted with 3 mL chloroform, followed by drying with a rotary evaporator and a nitrogen gas blowing concentrator. The AFB_1_ extract was dissolved with 100 μL trifluoroacetic acid and 200 μL n-hexane, and kept in the dark at 40 ℃ for 15 min. Subsequently, the derivative was dried under nitrogen gas blowing, and redissolved with 1 mL of acetonitrile -water (30:70; *v*/*v*). At last, the AFB_1_ derivative solution was filtered through a 0.22 μm filter, and quantified by using Aglient HPLC 1200 liquid-chromatograph (Agilent Technologies, Palo Alto, CA, USA) matched with a C18 column (Diamonsil^®^, 250 mm × 4.6 mm, 5 μm) conditioned at 30 °C. The mobile phase was 30% acetonitrile and 70% water at a flow rate of 1.0 mL/min. Then the signal of AFB_1_ was monitored with a fluorescent detector at 360/440 nm excitation/emission wavelengths. The AFB_1_ production were calculated according to the standard curve.

The content of norsolorinic acid (NOR) was quantitated as reported [42] with some modifications. The samples were mixed with 3 mL 9:1 methanol/1 N NaOH. NOR is a pH indicator and is purple at this pH, which allows for spectrophotometric measurement at 560 nm with a 96-well-titer ELISA plate reader (Thermo Fisher, Waltham, MA, USA).

### 5.3. Intracellular ROS Determination

ROS in mycelia was visualized by using a fluorescence microscope (Nikon, Japan) with a fluorescent probe of DCFH-DA (Sigma-Aldrich, St. Louis, MO, USA) described by [43] with slight modification. Spore suspension of *A. flavus* was incubated into YES with 1 mg/mL EGCG and cultured at 28 ℃ for 24, 36, 48 h. The mycelia were harvested by filtration, washed twice, and resuspended in PBS. The probes were loaded into the mycelia and incubated at 37 °C for 20 min. The treated mycelium was taken and the removal reagent was washed with PBS. Finally, mycelial resuspension in 100 μL PBS was directly transferred into the wells of 96-well plate directly for fluorescent photography. The relative intensity of fluorescence at 500 nm excited by 450 and 520 nm were time-course recorded by a microreader (Thermo Fisher Scientific, Waltham, MA, USA). And ROS distribution in mycelial tip was directly evaluated through fluorescence microscope.

### 5.4. RNA-Seq and RT-qPCR Validation

The *A. flavus* mycelia exposed to 1 mg/mL EGCG were taken and grounded in liquid nitrogen. RNA extractor (Trizol) kit (Sangon, Shanghai, China) was employed to extract total RNA according to the instructions. The integrity and quality of RNA were evaluated using 1% agarose gel and an Agilent Technologies 2100 Bioanalyzer. The cDNA library was constructed based on high-quality samples and then sequenced. The mRNA was separated by magnetic beads attached with oligonucleotide (dT). The isolated mRNA was cleaved into small fragments by divalent cations at high temperature. The cDNA was synthesized using these RNA fragments as templates. Appropriate fragments were picked as templates for the PCR amplification. Sequencing was performed by NovelBio Technology (Shanghai, China). Clean sequenced reads were obtained by filtering low quality reads and adaptor contamination, and were compared with the predicted transcripts of the *A. flavus* NRRL 3357 genome. The FPKM method was used to calculate and normalize the transcription levels of genes in *A. flavus*. The DEGs were identified as *p*-value< 0.05. DEGs were visualized by a volcano plot and then were subjected to GO enrichment analysis (Fisher’s Exact Test) and KEGG (Kyoto Encyclopedia of Genes and Genomes) pathway enrichment to identify which DEGs were dramatically enriched in GO terms or metabolic pathways.

Furthermore, the expression level of selected DEGs was confirmed by RT-qPCR. The primer sequences and gene names used in this study were displayed in Table 4. PCR program was settled as follows: 3 min denaturation at 95 °C followed by 35 cycles of 95 °C for 15 s and 60 °C for 40 s. The 2-ΔΔCT method was used to compute the expression of relative gene [44].

### 5.5. Western Blot Analysis

Spores (10^6^ cells/mL) were inoculated into YES with 1 mg/mL EGCG (without EGCG for the control), and incubated at 28 °C for 3 days. Mycelial biofilm formed on the surface was ground into fine powder in liquid nitrogen. The ground mycelia were addressed in RIPA (Beyotime Institute of Biotechnology, Jiangsu, China) lysis buffer for the total protein preparation. The amount of extracted protein was detected by using a BCA protein assay kit (Beyotime Institute of Biotechnology, Jiangsu, China). The derivative protein was exposed to 10% Sodium dodecyl SDS-PAGE gel and transferred to a polyvinylidene fluoride membrane (PVDF) (Millopore, Burlington, MA, USA). The membrane was sealed with 1× TBST solution containing 5% skim milk powder at room temperature for 1 h and then incubated overnight with primary antibody at 4 ℃. After washing with PBST solution for three times, the membranes were incubated with corresponding respective secondary antibody (1:2000, cell signaling) (Mouse anti-rabbit IgG-HRP, Santa Cruz Biotechnology, Dallas, TX, USA; Anti-mouse IgG; HRP-linked Antibody; Cell Signaling Technology, Danvers, MA, USA) for 2 h at room temperature. Protein gangs were visualized by using Chemi Doc-It^®^ 510 Imager (Upland, CA, USA), and the bands were quantitated by using Image J (Bio-Rad, Hercules, CA, USA). Alfa-tubulin (Sigma-Aldrich, St. Louis, MO, USA) was employed as an internal control [44,45].

### 5.6. Anti-Aflatoxigenic Evaluation of EGCG in Fermented Soybean Products

Fresh soybeans were boiled and ground which were purchased from a local supermarket. The soybeans (2 g) were placed in centrifuge tubes with 10 mL fungal suspension containing 10^6^ cells/mL. EGCG was added into the centrifuge tubes with the concentrations of 0, 1, 2, 4, 8 mg/mL, respectively. The tubes were incubated at 30 ℃ for 5 days. Then, AFB_1_ production in fermented soybean products were determined by HPLC, as described earlier.

### 5.7. Statistical Analyses

All experiments were conducted in triplicate. The results are presented as mean values with standard deviations. Statistical differences were assessed by Student’s *t*-test.

## Figures and Tables

**Figure 1 toxins-13-00693-f001:**
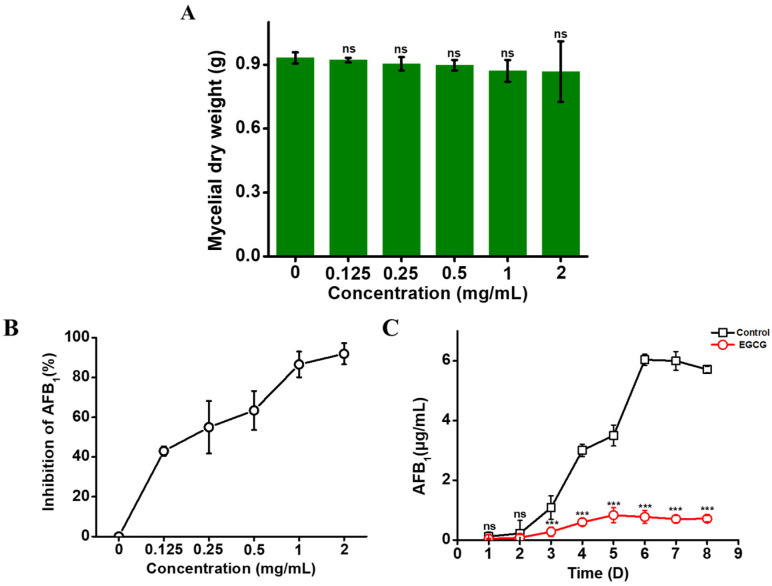
Effects of EGCG on mycelial growth and AFB_1_ production in *A. flavus*. (**A**) The dry weight of mycelia exposed to EGCG. (**B**) The AFB_1_ levels in YES (yeast extract with supplements) after treatment with EGCG at different concentrations for 5 days was determined by HPLC. (**C**) Constant monitoring of AFB_1_ biosynthesis in YES with 1 mg/mL EGCG for 8 days by HPLC. “ns” represents not significant; ***: *p* < 0.001.

**Figure 2 toxins-13-00693-f002:**
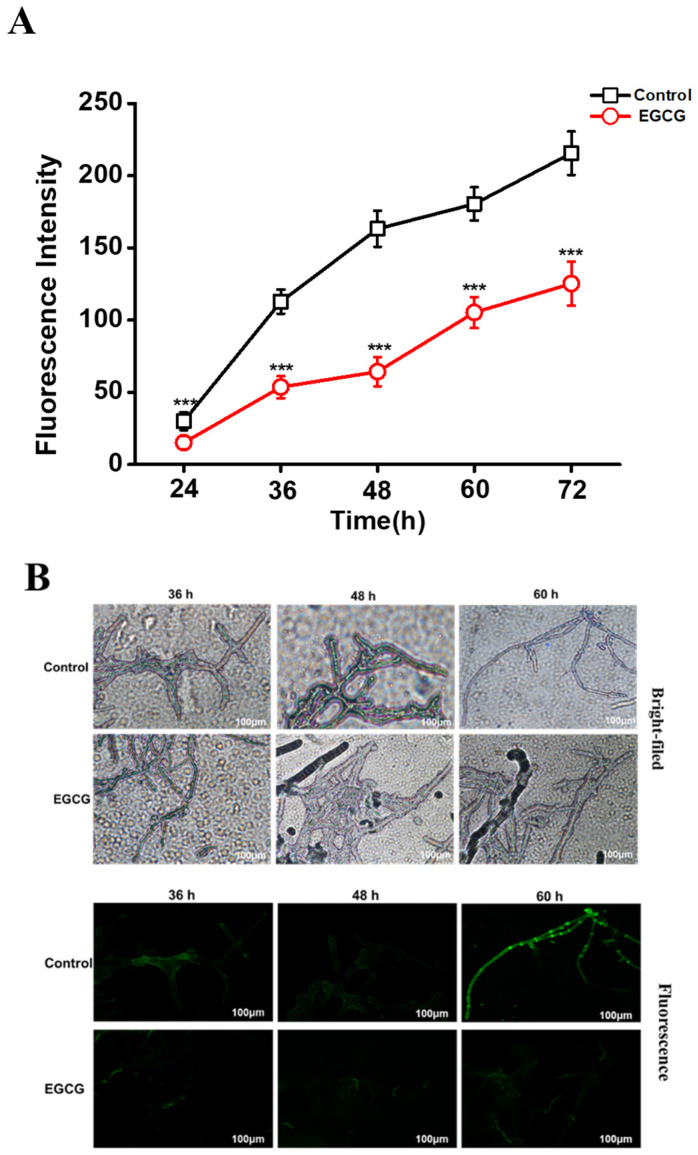
Influences of EGCG on mycelial ROS accumulation. (**A**) The fluorescence intensity was tested post the treatment with 1 mg/mL EGCG for 72 h by a microreader. (**B**) The mycelial bright-filed images and corresponding fluorescent images were obtained post treatment with 1 mg/mL EGCG for 36, 48 and 72 h by a fluorescence microscope. ***: *p* < 0.001.

**Figure 3 toxins-13-00693-f003:**
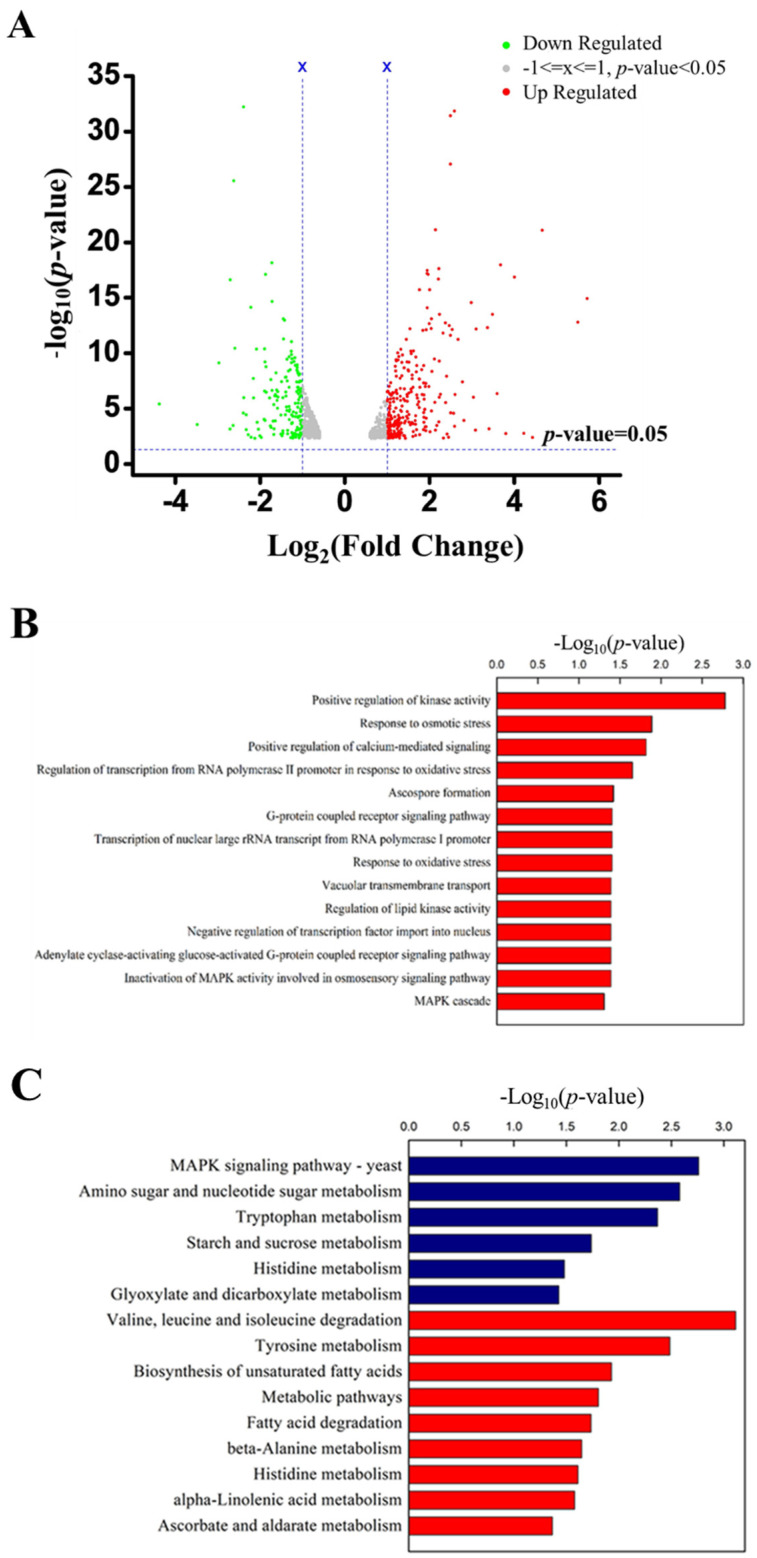
RNA-seq of *A. flavus* exposed to 1 mg/mL EGCG. (**A**) Volcano map describing the expression of different genes. (**B**) GO enrichment of down-regulated genes. (**C**) KEGG histogram of down-regulated DEGs. (**D**) Validation of RNA−seq data about the AFB_1_ biosynthesis down−regulated genes cluster by qRT−PCR. ***: *p* < 0.001.

**Figure 4 toxins-13-00693-f004:**
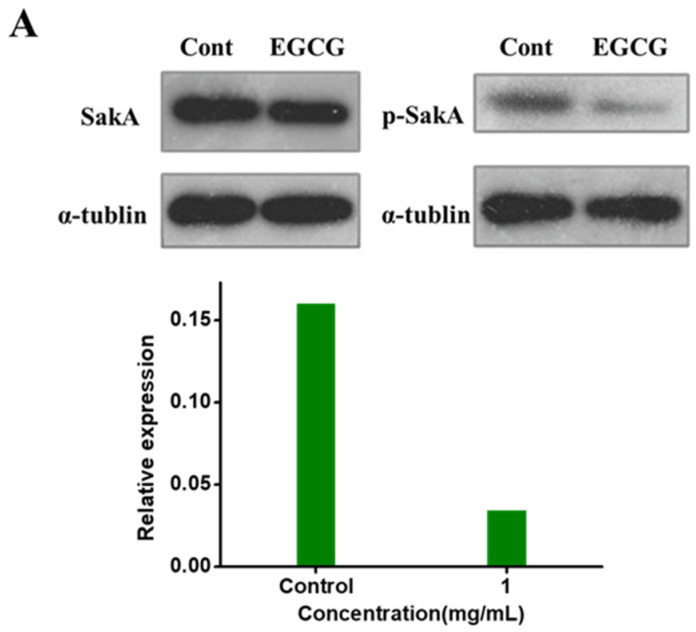
Influence of EGCG on SakA expression and phosphorylation in cytoplasm and nucleus. (**A**) The analysis of SakA phosphorylation and the protein expression of SakA in cytoplasm by Western blot. (**B**) The analysis of SakA phosphorylation and expression in nucleus by Western blot.

**Figure 5 toxins-13-00693-f005:**
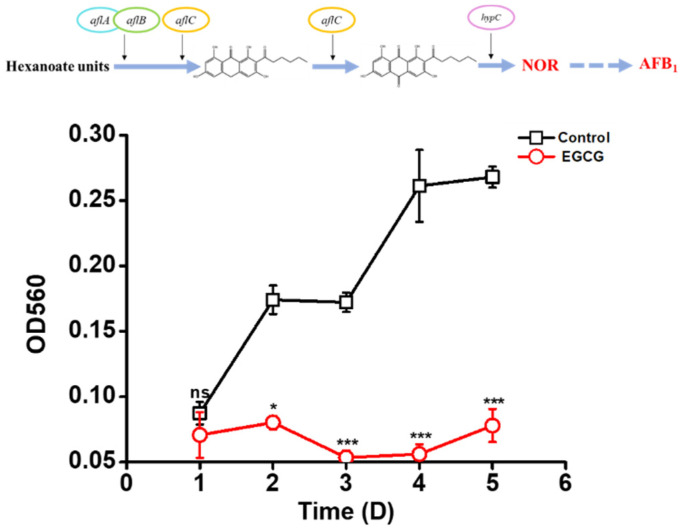
Effect of 1 mg/mL EGCG on the production of NOR in YES. NOR content was detected by optical adsorption at 560 nm with a microreader. “ns” means no significant; *: *p* < 0.05; ***: *p* < 0.001.

**Figure 6 toxins-13-00693-f006:**
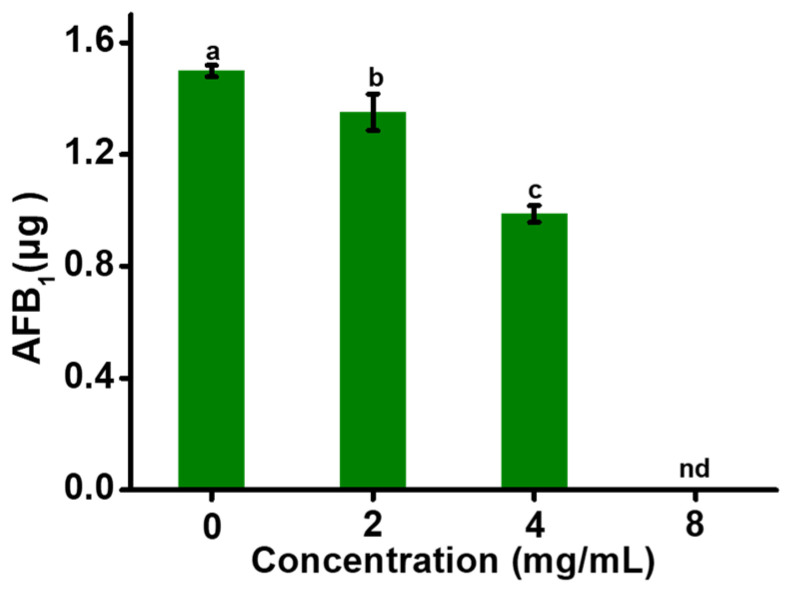
AFB_1_ amount in fermented soybean products was determined by HPLC. “nd” represents no detection, and different letters show a significant difference at *p* < 0.05.

**Figure 7 toxins-13-00693-f007:**
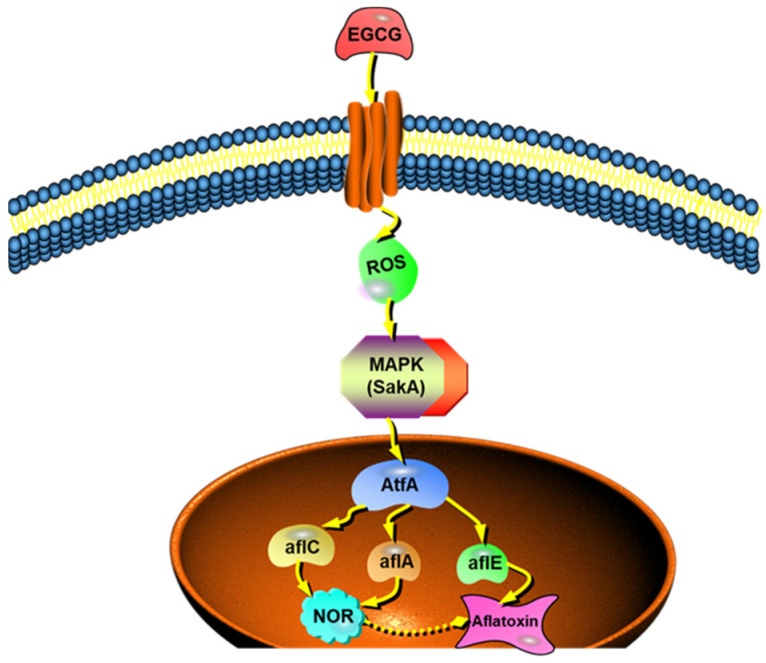
Hypothetical mechanism of EGCG inhibition of AFB_1_ biosynthesis in *A. flavus*. EGCG perturbed MAPK signaling pathway by decreasing the level of intracellular ROS. Simultaneously, the decrease of phosphorylated SakA level might reduce the tolerance to oxidative stress, leading to the activation of bZIP transcription factor AtfA in the stress signaling pathway. The decreased expression of genes in AFB_1_ cluster might then be the most down-stream response to the modulation by bZIP transcription factor.

**Table 1 toxins-13-00693-t001:** Transcriptional down-regulated genes and functions.

GOID	GOTerm	QueryID	Description	*p*-Value	Enrichment
GO:0033674	positive regulation of kinase activity	AFLA_034120	vacuole-associated enzyme activator complex component Vac14	0.001	0.09
GO:0033674	positive regulation of kinase activity	AFLA_046890	phospholipid metabolism enzyme regulator	0.001	24.59
GO:0006970	response to osmotic stress	AFLA_094490	protein phosphatase 2C	0.01	4.28
GO:0006970	response to osmotic stress	AFLA_136040	chitin synthase	0.01	4.28
GO:0006970	response to osmotic stress	AFLA_136030	chitin synthase ChsE	0.01	4.28
GO:0006970	response to osmotic stress	AFLA_013690	chitin synthase C	0.01	4.28
GO:0050850	positive regulation of calcium-mediated signaling	AFLA_051240	MAP kinase kinase (Mkk2) MAP	0.02	9.84
GO:0050850	positive regulation of calcium-mediated signaling	AFLA_068440	calcium channel subunit Cch1	0.02	9.84
GO:0043619	regulation of transcription from RNA polymerase II promoter in response to oxidative stress	AFLA_106830	sensor histidine kinase/response regulator Fos-1/TcsA	0.02	8.20
GO:0030437	ascospore formation	AFLA_137060	plasma membrane SNARE protein (Sec9)	0.04	2.67
GO:0030437	ascospore formation	AFLA_052430	G protein complex alpha subunit GpaB	0.04	2.67
GO:0030437	ascospore formation	AFLA_030580	C_2_H_2_ transcription factor PacC	0.04	2.67
GO:0007186	G-protein coupled receptor signaling pathway	AFLA_018540	G protein complex alpha subunit GpaB	0.04	6.15
GO:0007186	G-protein coupled receptor signaling pathway	AFLA_061620	a-pheromone receptor PreA	0.04	3.33
GO:0006979	response to oxidative stress	AFLA_106830	sensor histidine kinase/response regulator Fos-1/TcsA	0.04	3.07
GO:0006979	response to oxidative stress	AFLA_034380	spore-specific catalase CatA	0.04	3.07
GO:0006979	response to oxidative stress	AFLA_056170	mycelial catalase Cat1	0.04	3.07
GO:0000173	inactivation of MAPK activity involved in osmosensory signaling pathway	AFLA_094490	protein phosphatase 2C	0.04	24.59
GO:0010619	adenylate cyclase-activating glucose-activated G-protein coupled receptor signaling pathway	AFLA_018540	G protein complex alpha subunit GpaB	0.04	24.59
GO:0042992	negative regulation of transcription factor import into nucleus	AFLA_032170	protein serine/threonine kinase (Ran1)	0.04	24.59
GO:0043550	regulation of lipid kinase activity	AFLA_034120	vacuole-associated enzyme activator complex component Vac14	0.04	24.59
GO:0035970	peptidyl-threonine dephosphorylation	AFLA_094490	protein phosphatase 2C	0.04	24.59
GO:0071619	phosphorylation of RNA polymerase II C-terminal domain serine 2 residues	AFLA_005900	protein kinase	0.04	24.59
GO:0034486	vacuolar transmembrane transport	AFLA_112050	cation chloride cotransporter	0.04	24.59

**Table 2 toxins-13-00693-t002:** The numbers of DEGs in each pathway.

Pathway ID	Regulated	Pathway Term	Gene Number
PATH:04011	Down	MAPK signaling pathway-yeast	9
PATH:00520	Down	Amino sugar and nucleotide sugar metabolism	9
PATH:00380	Down	Tryptophan metabolism	7
PATH:00500	Down	Starch and sucrose metabolism	8
PATH:00340	Down	Histidine metabolism	3
PATH:00630	Down	Glyoxylate and dicarboxylate metabolism	5
PATH:00280	Up	Valine, leucine and isoleucine degradation	7
PATH:00350	Up	Tyrosine metabolism	8
PATH:01040	Up	Biosynthesis of unsaturated fatty acids	4
PATH:01100	Up	Metabolic pathways	53
PATH:00071	Up	Fatty acid degradation	5
PATH:00410	Up	beta-Alanine metabolism	4
PATH:00340	Up	Histidine metabolism	3
PATH:00592	Up	alpha-Linolenic acid metabolism	2
PATH:00053	Up	Ascorbate and aldarate metabolism	2

**Table 3 toxins-13-00693-t003:** Influence of EGCG on the transcriptions of gene cluster responsible for aflatoxin production.

Gene ID	Gene Name and Product	Style
AFLA_139310	*aflE*/*norA*/*aad*/*adh2*/NOR (norsolorinic acid reductase)	down ^a^
AFLA_139320	*aflJ*/*estA*/esterase	down
AFLA_139360	*aflR*/*apa2*/*afl2*/C6 transcription factor	down
AFLA_139380	*aflA*/*fas2*/*hexA*/fatty acid synthase alpha subunit	down ^a^
AFLA_139410	*aflC*/*pksA*/*pksL1*/polyketide synthase	down ^a^

^a^ shows a significant difference at *p* < 0.05.

**Table 4 toxins-13-00693-t004:** The list of primers for qRT-PCR.

Gene	Forward Primer (5′-3′)	Reverse Primer (5′-3′)
*aflA*	CGTCACGCTCTATACAATTTGCT	ATCCGATAAAGTTGCCTAGTTCC
*aflC*	TGTCAGACCACAAACGCACCT	CATCTCACAGAACGCCCTCAA
*aflE*	CTATCATCTAGCGCCGGTGT	CCATCTTTCGCTATCGCCTCC
*aflJ*	CTGCGTTGCTACACTCCCC	ATCACGCGGCAGAAACCATC
*aflR*	GCAGTCAATGGAACACGGAAAC	CCTGAAACGGTGGTAGTGGG
*atfA*	AAACTGAAGACTCCCAGGCGC	AAACTGAAGACTCCCAGGCGC
*actin*	AGGACTCTTATGTCGGTGATG	CGGTTGGACTTAGGGTTG

## Data Availability

No new data were created or analyzed in this study. Data sharing is not applicable to this article.

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
