# Peer review of "EGCG Alleviates Oxidative Stress and Inhibits Aflatoxin B1 Biosynthesis via MAPK Signaling Pathway"

_toxins, 2021, doi:10.3390/toxins13100693_

Round 1

Reviewer 1 Report

Following comments must be considered for improving the quality of this manuscript. After these changes are made, the manuscript should be considered for publication.

  1. Line 16 – suggested to use some other word and not forcefully
  2. There are minor grammatical errors throughout the manuscript so the manuscript should be thoroughly edited for clarity
  3. In Fig. 1A, the significance should be reported with * or ns, which is the accepted convention
  4. Are the images displayed in Fig. 2B normalized for fluorescence intensity?
  5. In line 97, DEGs should be written in full form. Generally advised to use full forms and then abbreviations in the follow ups
  6. It is possible that p-SakA turnover in the nucleus could be an indirect consequence of the ROS so have the authors tried other upstream and downstream markers in the MAPK pathway to bolster their claims?
  7. Figure 7. Is it hypothetical or proposed model? Please be specific

Author Response

Dear Reviewer,

Thank you for your kind and patient work on our manuscript entitled as “EGCG alleviates oxidative stress and inhibits aflatoxin B1 biosynthesis via MAPK signaling pathway” (Toxins-1369502).

According to your suggestions and the comments, we have checked the manuscript and made careful revisions including revising some inappropriate expressions and grammatical errors, updating figures with statistical significance. We are very grateful to your comments which greatly improved the quality of our manuscript. We are looking forward to your response soon. We would thank you again for your comments.

Yours sincerely

Reviewer 2 Report

The manuscript demonstrates the effect of epigallocatecgin-3-gallate (EGCG), the major constituent of green tea polyphenol in ameliorating oxidative stress by inhibiting aflatoxin B1 biosynthesis. The authors have conducted the study demonstrating that EGCG is effective in reducing oxidative stress via MAPK signaling pathways. The authors have conducted NGS to identify pathways and target molecules regulating this response. Overall, the work is interesting however, a similar article published in this area (Toxins (Basel). 2020 Dec 6;12(12):777) where the authors have shown degradation and detoxification of aflatoxin B1 by tea polyphenols, lowering the innovation. There are few minors which needs attention.

  1. Figure 1A, what does a stands for in each bar?
  2. Figure 1C, 2A, statistical significance is missing.
  3. Figure 3, authors have provided the relative expression data by q-RT-PCR analysis, however statistical analysis is missing.
  4. The authors should conduct additional WB to show which MAPK family member protein expression is altered by EGCG.
  5. Figure 5 and 6, statistical analysis is missing.

Author Response

Dear Reviewer,

Thank you for your kind and patient work on our manuscript entitled as “EGCG alleviates oxidative stress and inhibits aflatoxin B1 biosynthesis via MAPK signaling pathway” (Toxins-1369502).

According to your suggestions and the comments, we have checked the manuscript and made careful revisions including revising some inappropriate expressions, updating figures with statistical analysis. We are very grateful to your comments which greatly improved the quality of our manuscript. We are looking forward to your response soon. We would thank you again for your comments.

Yours sincerely

Round 2

Reviewer 2 Report

The authors have satisfactorily addressed the comments raised. No further comments.

Author Response

Dear Reviewer,

    Thank you for your kind and patient work on our manuscript. We are very grateful to your comments which greatly improved the quality of our manuscript. 

Yours sincerely

Liangbin Hu

School of Food and Biological Engineering

Shaanxi University of Science and Technology, Xi’an, China